# Water Effects on Molecular Adsorption of Poly(N-vinyl-2-pyrrolidone) on Cellulose Nanocrystals Surfaces: Molecular Dynamics Simulations

**DOI:** 10.3390/ma12132155

**Published:** 2019-07-04

**Authors:** Darya Gurina, Oleg Surov, Marina Voronova, Anatoly Zakharov, Mikhail Kiselev

**Affiliations:** G.A. Krestov Institute of Solution Chemistry of the Russian Academy of Sciences, 1 Akademicheskaya St., Ivanovo 153045, Russia

**Keywords:** cellulose nanocrystals, polyvinylpyrrolidone, adsorption, self-assembly, molecular dynamics

## Abstract

Models of interaction between a poly(N-vinyl-2-pyrrolidone) macromolecule and a fragment of I_β_-cellulose were built in a vacuum and water environment. The models were made to interpret the mechanism of interaction of the polymer and cellulose nanocrystals by the classical molecular dynamics method. The structural behavior of a poly(N-vinyl-2-pyrrolidone) macromolecule in water has been studied in terms of the radius of gyration, atom–atom radial distribution functions and number of hydrogen bonds. It was found that the polymer has a high affinity with the solvent and each monomer unit has on average 0.5 hydrogen bonds. The structural and energy characteristics of the polymer adsorption were investigated at different initial positions of the poly(N-vinyl-2-pyrrolidone) macromolecule relative to the cellulose fragment. It was observed that the polymer macromolecule was mainly adsorbed on the cellulose fragment in the globular form. Moreover, in the solvent the interaction of poly(N-vinyl-2-pyrrolidone) with the cellulose hydrophobic surface was stronger than that with the hydrophilic one. This study will show that the presence of water makes the interaction between the polymer and cellulose weaker than in a vacuum, and the polymer and cellulose mainly interact through their solvation shells.

## 1. Introduction

Cellulose is the most abundant available renewable polymer resource. Cellulose is well known as a raw material used in a wide spectrum of materials and products. Cellulose fibrils are structural entities formed through cellulose biogenesis, stabilized by van der Waals forces and hydrogen bonds. The fibrils can be separated from a cellulose source into amorphous and crystalline components by enzymatic, chemical, mechanical processes or their combination, yielding cellulose nanofibrils (CNF) or cellulose nanocrystals (CNC).

CNC are the subject of a plethora of studies in the development of coating components, rheology modifiers, stabilizers of multiphase systems, optical devices, and nanocomposites reinforcement, owing to the CNC nanoscale dimensions, light weight, morphology, and self-assembly behavior [1,2,3,4]. The unique physical and chemical properties of CNC, such as high stiffness and strength, biodegradability, low density and high aspect ratio, are discussed in a number of reviews [5,6,7]. CNC have attracted a lot of attention in the materials community and were recently used to engineer new materials. For instance, combinations of synthetic water-soluble polymers—polyvinyl alcohol (PVA), polyethylene oxide (PEO), polyvinylpyrrolidone (PVP)—with CNC were investigated for some biomedical and pharmaceutical applications, as solid polymer electrolytes [8,9,10,11,12,13], etc.

As in a typical colloidal system, the properties of CNC aqueous suspensions, for example, rheology, stability and optical characteristics, depend on the surface area, shape, and size of the CNC particles. At relatively high concentrations, CNC particles can self-assemble into a chiral nematic liquid crystalline phase, a property that is widely used in the production of functional films [14].

Recently, we have for the first time discovered the ability of CNC to self-assemble with PVP assistance producing uniform CNC aggregates with a high aspect ratio (length/width) [15]. The considered possible model of the PVP-assisted CNC self-assembly supposes that adsorbed PVP molecules block the lateral bonds between the CNC particles, and promote such a self-assembly. PVP, a water-soluble, non-ionic amorphous polymer is widely used in nanoparticle synthesis [16]. Due to the amphiphilic nature, PVP can affect morphology and growth of nanoparticles by ensuring their solubility in various solvents, discriminatory surface stabilization, controlled crystal growth, playing the role of a shape-control agent, and facilitating the growth of specific crystal faces while preventing others [17]. For instance, PVP can get strongly bound to the (100) facets of Ag, facilitating Ag nanowire growth along the (111) direction [18].

Adsorption on surfaces from a solution phase is widespread in many materials applications. As mentioned above, nanocrystals grown in a solution can be stabilized by adsorbed molecules which control the nanocrystal shape. In this work, we use classical molecular dynamics simulations to quantify vacuum and solution-phase (water) interactions of PVP with different facets of CNC—a system studied experimentally for solution-phase adsorption of PVP on CNC. The aim of this research is to reveal and prove the appropriate mechanism of PVP adsorption onto different facets of CNC particles, facilitating the PVP-assisted CNC self-assembly. Despite the fact that the properties of complex systems containing PVP have been successfully investigated with atomistic-scale simulations before [17,18], the interactions between PVP and cellulose in water have, to the best of our knowledge, never been investigated by molecular modeling. The present study attempts to fill this gap and provide detailed information about the behavior of PVP in water and its interaction with cellulose I_β_ in the absence and presence of the solvent. Comparing predictions and experimental data will help to contribute toward to a better understanding of the properties of such complex systems which depend not only on the molecular structure features but also on the specific interactions of the system components.

## 2. Computational Details

Classical molecular dynamics simulations were carried out using a GPU-accelerated (Graphics Processing Unit) Gromacs-5.0.7 software package [19]. Molecular graphics and visualization were performed using VMD 1.8.6 [20]. The molecular dynamics (MD) simulations were carried out for the NVT ensemble (constant number of particles N, volume V and temperature T). The reference temperature of 298 K was kept constant using a Nose–Hoover thermostat [21,22] with the coupling constant τ = 0.1 ps. Periodic boundary conditions were applied to all three directions of the simulated cubic box. The Verlet algorithm [23] was adopted to integrate the equations of motion. The modified Ewald summation method [24,25] was used to account for the corrections of the long-range electrostatic interactions with a cutoff radius of 1.5 nm, which was also the cutoff value for the Van Der Waals (VDW) interactions. All the bond length constraints were implemented using the LINCS algorithm (LI Near Constraint Solver) [26]. In previous studies a few different potential models were used for PVP. Among them, potential models based on the AMBER and COMPASS force fields were employed for simulation of the amorphous phase of PVP [27,28,29,30,31], and the united atoms model [18] based on GROMOSG53a6 parameters was used for studying PVP-coated silver nanoparticles. In our work for PVP (Figure 1a) we utilized the united atoms potential model from [18]. In order to make sure that this model is suitable for modeling PVP in water we additionally carried out simulation with potential based on OPLS (Optimized Potential for Liquid Simulations) force field parameters [32]. The initial structure of a PVP macromolecule containing 60 monomer units (with the molecular weight of 6668.646 g/mol) was constructed by means of Avogadro [33]. This number of monomer units is sufficient to observe the conformational transitions of the polymer and to obtain sufficient statistical data to calculate quantitative characteristics. For cellulose we used GROMOS54a7 force field parameters [34]. The initial structure of the cellulose fragment (CEL) was built based on experimental crystallographic data [35] by a toolkit named Cellulose Builder [36].The model of the I_β_ cellulose consisted of 14 glucan chains, and the degree of polymerization of each chain was 10 (Figure 1). The number of chains and degree of polymerization provide a sufficient surface area for effective interactions with the polymer. The systems were solvated by explicit water molecules using the SPC/E (Extended Simple Point Charge) model [37]. The size of the water box was chosen to ensure that the systems had at least a 20 Å solvation shell in all the directions. After a system energy minimization, we equilibrated the system for 0.5 ns in the NVT ensemble. The production run simulations were performed for 20 ns with a time step of 1 fs. The data for the analysis were collected every 0.1 ps. Eight systems were simulated and the simulation details are listed in Table 1. Initial systems 1 and 2 were constructed placing, respectively, 1 or 6 PVP molecules with a nearly extended conformation into a cubic box with 32,463 water molecules. Initial systems 3 and 5 were built by inserting a cellulose fragment into the center of the cubic box. Then a PVP macromolecule was placed near the hydrophobic (System 3) or hydrophilic (System 5) side of the cellulose fragment followed by addition of 32,463 water molecules (Figure 2). The initial configurations of Systems 4 and 6 were obtained by removing the water molecules from Systems 3 and 5, respectively. Using such a way of construction of the initial PVP-cellulose system (Systems 4 and 6) we tried to imitate the process of the composite PVP/CNC drying. The initial configurations of Systems 0 and 7 consist of a PVP molecule in an extended conformation in a vacuum and the cellulose fragment in the cubic box with 32,463 water molecules, respectively.

## 3. Results and Discussion

In order to understand the structural behavior of the polymer in the solvent, we calculated the radius of gyration (*R_g_*) which is one of the important quantities in conformational statistics of polymer chains depending on the molecular weight of the macromolecule, on its constitution (whether or not and how it is branched), and on the extent to which it is swollen by the solvent. The radius of gyration is a measure of the effective size of a polymer and shows the radius distribution of polymeric chains in the radial chain direction [38]. Therefore, it was used to quantify the PVP folding degree. The radius of gyration was calculated by the following equation:(1)$$R_{g}={\left(\frac{{\sum_{i}\left\|r_{i}\right\|}^{2}m_{i}}{\sum_{i}m_{i}}\right)}^{1/2}$$
where *m_i_* is the mass of site *i* and *r_i_* is the position of site *i* relative to the center of mass of the molecule. Figure 3 shows the simulation time dependence of the PVP radius of gyration in a vacuum (system 0) and in water (system 1) for two PVP potential models based on GROMOS and OPLS force fields as mentioned in Section 2. As can be seen, in water, there is a gradual decrease in the gyration radius during the first 7 ns and 14 ns for GROMOS and OPLS force field, respectively, and then *R_g_* fluctuates around a constant value (i.e., the polymer chain condenses onto itself into a dense conformation which is called a polymer globule). The process of a PVP molecule folding in water is shown in Figure 4. The same process is observed in system 2 with six polymer molecules and the average value of *R_g_* is about 1.184 nm (Table 2). In the absence of the polar solvent, the polymer folding is more than 50 times faster. Since the behavior of the polymer in water is the same for two PVP models, simulations of systems 3–6 were carried out only using the potential developed in [18].

Radial distribution function RDF reflects the law of specific interactions and can be used to investigate the polymer solvation mechanism in water. The RDF results of the O6, C7, C8, C1, C2, C3, N atoms of PVP and oxygen Ow and hydrogen Hw atoms of water for system 1 and system 2 (Table 1) are shown in Figure 5. The presence of a highly polar five-membered lactam cycle in PVP is responsible for its hydrophilic character while the hydrophobicity is due to the carbon atoms of its chain. It is clearly seen that the PVP oxygen atom O6 is more strongly solvated by water molecules than the C7 and C8 atoms of the PVP hydrophobic chain. The RDF analysis shows that the water molecules are mainly located around the C1, C2, C3 atoms of the ring rather than the N one. It indicates that the nitrogen atom does not participate in the hydration process although it has a lone p-electron pair, which is a potential hydrogen acceptor. This result is in agreement with the IR spectroscopic data [39]. In contrast, a narrow peak on the RDF from 0.15 to 0.26 nm related to O6–Hw interactions can be attributed to hydrogen bonding. It is well known that PVP absorbs water well [40], probably, due to the PVP ability to take part in hydrogen bonding with the solvent molecules. To quantify a specific interaction between the polymer and solvent, we calculated the average number of hydrogen bonds per PVP monomer unit (<n_HB_>) (Figure 6, Table 2). The hydrogen bonds were counted based on the geometrical criteria of a donor–acceptor distance of0.35 nm, and an acceptor–donor–hydrogen angle of 30°.

It is clearly seen that the time dependence of *R_g_* correlates with the time dependence of the number of polymer-solvent hydrogen bonds (HBs) (Figure 3 and Figure 6). PVP transition from the expanded chain conformation to the globular form causes dehydration. If at the beginning of the simulation almost every center of hydrogen bonding is involved in the formation of HBs with water molecules, then after 7 ns the average number of HBs per oxygen atom of PVP is about 0.55. An increase in the PVP concentration in the solution results in a decrease in the <n_HB_> PVP–water (Table 2, system 2). It is due to the interaction of the PVP macromolecules with each other and, respectively, decrease in available hydrogen bonding centers.

Ideally, CNC are rod-like highly crystalline anisotropic particles with a high aspect ratio (15–20 nm in width, 50–300 nm in length). It is usually believed that the CNC derived from higher plant cell wall cellulose consist of cellulose chains arranged in an I_β_ monoclinic crystal structure and have a square cross-section with (110) and (1-10) terminating hydrophilic surfaces. Parallel stacking of the cellulose chains in one so-called ‘‘hydrogen bonding’’ plane forms hydrophobic (200) facets of CNC. Outside this ideal CNC structure, there is considerable variability in the CNC particle shape, for example, with a rectangular-shaped, parallelogram-shaped or hexagonal-shaped cross-section [41,42]. The shape of the CNC cross-section determines the surface ratio of hydrophilic and hydrophobic facets of the CNC particles and may affect their colloid and interfacial behavior [14,43,44]. In order to understand the adsorption process of PVP on CNC we have considered two models. In the first case (system 3), the PVP macromolecule in a globular conformation was placed near the hydrophobic surface of the cellulose fragment (CEL) (Figure 2). In the second case (system 5), the PVP molecule was placed near the hydrophilic surface. The time evolution of the number of contacts (Figure 7) between any pair of atoms from the PVP and CEL within a given distance (0.5 nm) and the average minimum distance between any pair of atoms (Table 2) shows that the process of PVP adsorption is slightly different for systems 3 and 5. In system 3, the number of close contacts between the PVP and CEL atoms gradually increases and after 3 ns reaches a constant value indicating the end of the adsorption process. The average number of close contacts (<N_C_>) between the CEL and adsorbed polymer is about 460, while in system 4, where water is removed, the average number of PVP–CEL contacts is four times higher. Evidently, PVP in a vacuum is adsorbed on CEL in a flattened expanded conformation while in a water environment it is adsorbed as a compact globular structure (considering evolution of *R_g_*, <N_C_> and <r_min_> values in Table 2). The same is observed for system 5; however, the number of PVP contacts with the hydrophilic surface of CEL is noticeably lower. Apparently, the polymer interacts with CEL indirectly through the water molecules. It should be noted that the experimental observation of the authors [45] showing easy PVP adsorption and desorption on microcrystalline cellulose in an aqueous solution may indicate that the polymer is adsorbed in the globular form, which is in agreement with our results. A visualization of the trajectories shows that in a vacuum PVP is attracted by both hydrophobic and hydrophilic surfaces of the cellulose fragment more quickly than in the solution. This is due to the electrostatic interactions between the negatively charged oxygen (−0.30 e.c.) and nitrogen (−0.25 e.c.) atoms of the pyrrolidone ring and the positively charged hydrogen (+0.42 e.c.) atoms of the cellulose hydroxyl groups which are much stronger in a vacuum than in water.

The average number of PVP–CEL hydrogen bonds (<n_HB_>) per PVP unit as well as HBs per glucose unit between cellulose and water, and internal CEL–CEL hydrogen bonds in the simulations, is shown in Table 2. Despite the fact that in systems 3 and 4 the surface of the cellulose fragment has a dominant hydrophobic character, there are a few hydrogen bonds between the polymer and the cellulose. In a water environment the average number of hydrogen bonds (<n_HB_>) PVP–CEL is 1.1 (system 3), while in a vacuum the <n_HB_> increases up to 3.5 (system 4). In system 5, PVP forms 0.5 HBs with the cellulose hydrophilic surface on average, and the number of hydrogen bonds between the PVP and water is bigger than in system 3. This fact also confirms our hypothesis that the polymer and cellulose mainly interact through their solvation shells (Figure 8). Moreover, a visual inspection of the trajectories shows that some water molecules form HBs with both PVP and cellulose simultaneously (Figure 8).

To quantify the adsorption energy in the gas phase and in the solvent, the energy of the interaction between the systems components was calculated. The interaction energy (Δ*E*) was obtained by subtracting the energies of the isolated components (*E_i_*) from the total energy of the complete system (*E_tot_*) [46,47,48,49]:(2)$$\Delta E=E_{tot}-\sum E_{i}$$

The interaction energies of all the systems with and without water are listed in Table 3.

As one can see, the energies of the PVP–water and CEL–water interaction are negative, showing a high affinity of the components with the solvent, which is consistent with their ability to form a sufficiently large number of hydrogen bonds with water molecules (Table 2). The interaction energy between the polymer molecule and two surfaces of the cellulose fragment in vacuum are also negative, and for the hydrophilic surface this value is about 200 kJ·mol^−1^ bigger than for the hydrophobic one, which is probably due to the PVP ability to form more HBs with the surface of the former than with that of the latter (Table 2). Comparing the energies of the PVP–water and CEL–water interactions with the PVP–CEL interaction in a vacuum, their differences in magnitude are significant (i.e., the PVP and cellulose fragment are more likely to interact with the solvent rather than with each other). It should be noted that in an aqueous environment, the total energy of the system (*E_tot_*) is a sum of energy contributions not only of the components but also of the PVP–water, CEL–water and PVP–CEL interaction energies. The gain in energy of formation of the polymer-cellulose complex in water is smaller than in a vacuum. This indicates that water molecules weaken the interaction between the PVP and cellulose. This is in line with the data discussed above on the average number of close contacts, <n_HB_>, <r_min_> and *R_g_*. Moreover, this observation agrees with our experimental results which show that the particle size of the CNC/PVP nanocomposites re-dispersed in water is the same as that of the CNC before PVP adsorption [15].

## 4. Conclusions

Classical molecular dynamics simulations were used to describe the complex physical phenomenon of PVP macromolecule adsorption on a cellulose I_β_ fragment (CEL) to quantify vacuum and aqueous phase interactions of the PVP with different facets of cellulose nanocrystals. The paper considers the interaction of PVP with both types of cellulose surfaces—hydrophobic and hydrophilic ones. Beginning the modeling from the expanded chain conformation of PVP surrounded by water molecules we observed a gradual transformation of the PVP into the globular form accompanied by dehydration (i.e., by a decrease in the number of water molecules interacting with the PVP through hydrogen bonding). In such a globular form PVP interacts with the cellulose fragment in water, while in a vacuum PVP is adsorbed in a flattened conformation due to the strong electrostatic interactions between the negatively charged oxygen and nitrogen atoms of the pyrrolidone ring and the positively charged hydrogen atoms of the cellulose hydroxyl groups. Thus, the solvent plays a crucial role in the interaction between PVP and CEL. In the solution interactions between the polymer and water, and CEL and water are significantly stronger than the interaction between PVP and CEL. Moreover, in the aqueous media water molecules weaken the interaction between PVP and CEL in comparison with those in a vacuum. In case of the hydrophilic side of cellulose this effect is more pronounced due to the high ability of the cellulose hydroxyl groups to form hydrogen bonds with water molecules. Despite the fact that there are some limitations of molecular dynamics simulations in terms of time and length scales (because of the high polymer molecular weight), a comparison between the predictions and experimental data for the ternary system PVP–CEL–water shows that atomistic modeling is a powerful tool, which improves the understanding of properties of complex systems depending on the features of the molecular structure and specific interactions of the system components.

## Figures and Tables

**Figure 1 materials-12-02155-f001:**
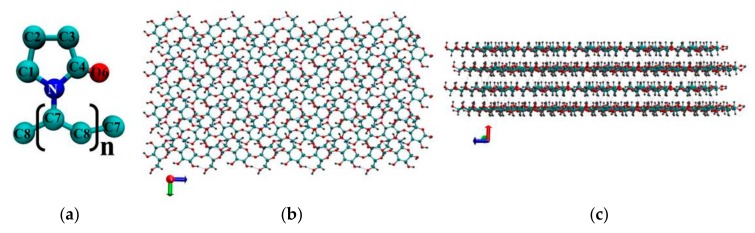
Atom numbering (without hydrogen atoms) in a monomer unit of polyvinylpyrrolidone (PVP) (**a**), hydrophobic surface of the cellulose fragment (**b**), hydrophilic surface of the cellulose fragment (**c**). The red spheres denote the oxygen atoms, the gray ones represent the hydrogen atoms, the cyan ones represent the carbon atoms, and the blue sphere represents the nitrogen atom.

**Figure 2 materials-12-02155-f002:**
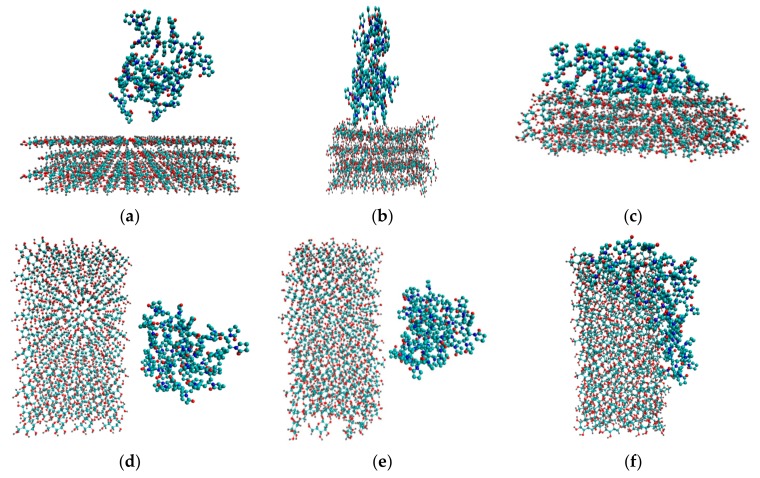
Initial configurations of system 3 (**a**) and system 5 (**d**). Snapshots of the last frames of system 3 (**b**) and system 5 (**e**), system 4 (**c**) and system 6 (**f**). All the water molecules were deleted to highlight the polymer and the cellulose fragment.

**Figure 3 materials-12-02155-f003:**
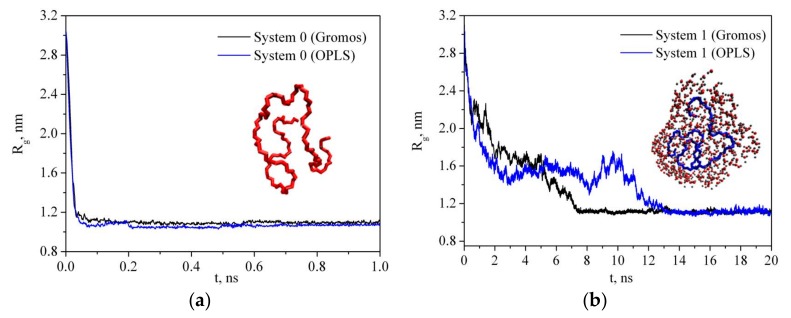
Time dependence of the radius of gyration of PVP in a vacuum (**a**) and in water (**b**).

**Figure 4 materials-12-02155-f004:**
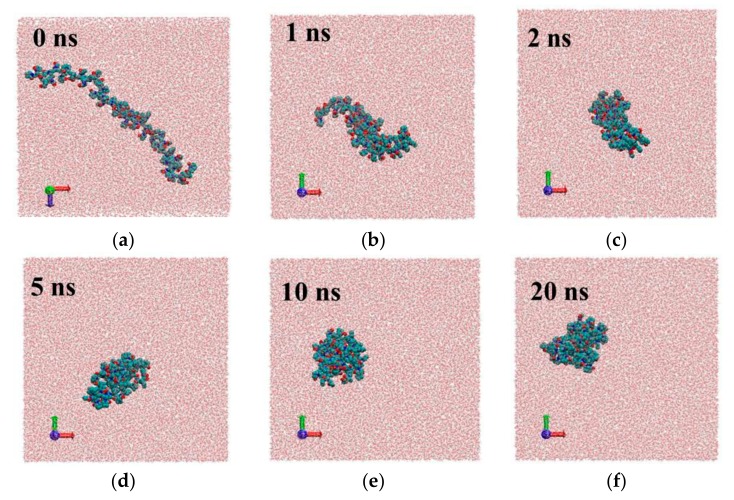
Snap shots of different frames for system1 with PVP potential based on GROMOS force field parameters [18]: 0 ns (**a**), 1ns (**b**), 2 ns (**c**), 5 ns (**d**), 10 ns (**e**), 20 ns (**f**).

**Figure 5 materials-12-02155-f005:**
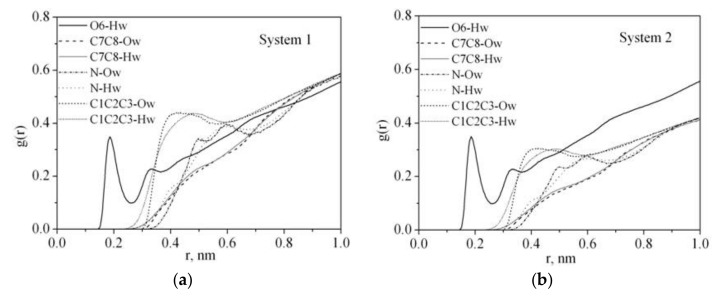
Atom–atom radial distribution functions for PVP in water: (**a**) system 1, (**b**) system 2.

**Figure 6 materials-12-02155-f006:**
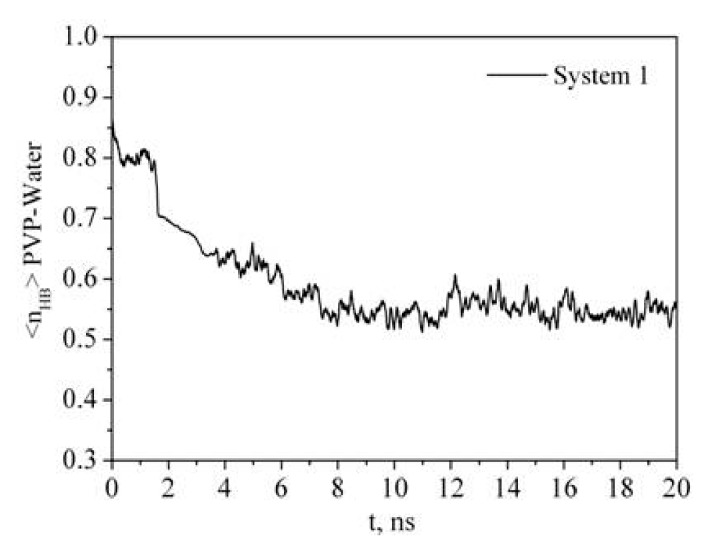
Time dependence of the number of PVP–water hydrogen bonds (<n_HB_>) per PVP monomer unit.

**Figure 7 materials-12-02155-f007:**
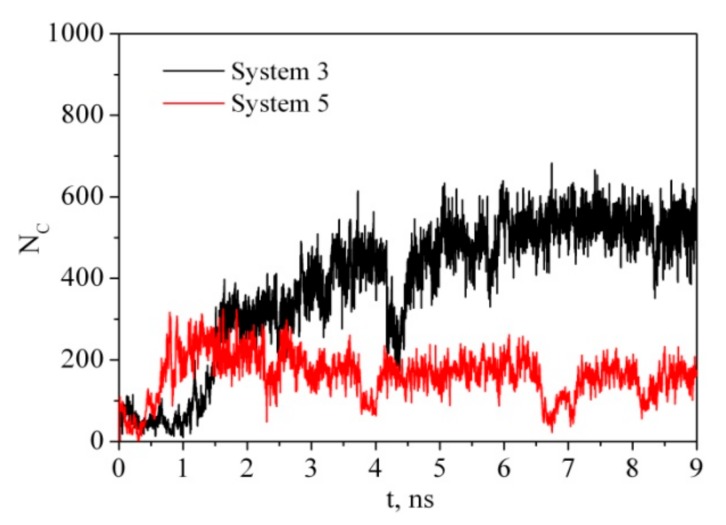
Time dependence of the number of contacts between any pair of atoms from PVP and CEL within a given distance of 0.5 nm.

**Figure 8 materials-12-02155-f008:**
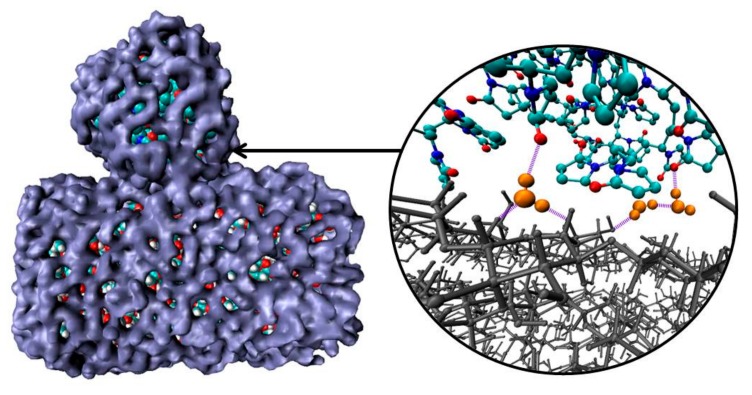
Snapshots of solvation shells (blue) of PVP and cellulose; hydrogen bonds (violet) forming by water molecules (orange) with PVP and cellulose (gray). Other water molecules have been deleted for clarity to highlight the polymer and the cellulose fragment interactions.

**Table 1 materials-12-02155-t001:** Details of different systems: number of PVP, cellulose fragment (CEL) and water molecules.

Systems	N(PVP)	N(CEL)	N(Water)
System 0 vacuum	1	-	-
System 1	1	-	32,463
System 2	6	-	32,463
System 3 (top)	1	1	32,463
System 4 (top) vacuum	1	1	-
System 5 (side)	1	1	32,463
System 6 (side) vacuum	1	1	-
System 7	-	1	32,463

**Table 2 materials-12-02155-t002:** The average number of hydrogen bonds (<n_HB_>) PVP–water (per PVP molecule and monomer unit), PVP–CEL (per PVP molecule), CEL–water and CEL–CEL (per CEL monomer unit), water–water (per water molecule), the radius of gyration (*R_g_*, nm), the average number of close contacts between any pair of atoms from PVP and CEL within the limits of 0.5 nm (<N_C_>), the average minimum distance between any pair of atoms from PVP and CEL within the bounds of 0.5 nm (<r_min_>, nm).

System	<n_HB_>					*R_g_* nm	<N_C_>	<r_min_> nm
	PVP–Water	PVP–CEL	CEL–Water	CEL–CEL	Water–Water			
0						1.149		
1	32.7 (0.55)				3.59	1.108		
2	25.2 (0.42)					1.184		
3	30.1 (0.50)	1.1	2.74	3.72	3.58	1.111	460	0.206
4		3.5		4.4		1.285	1851	0.173
5	35.8 (0.60)	0.5	2.75	3.78	3.58	1.259	170	0.247
6		5.1		4.4		1.524	1917	0.170
7			3.14	3.41	3.58			

**Table 3 materials-12-02155-t003:** Total energies of the systems (*E_tot_*), single point energies of the components and PVP–CEL complex (*E*_PVP_, *E*_CEL_, *E*_Water_, *E*_PVP–CEL_), total interaction energies between the systems components (Δ*E_tot_*), interaction energies between the PVP and CEL (Δ*E*_PVP–CEL_). Energy unit: kJ·mol^−1^.

System	*E_tot_*	*E* _PVP_	*E* _CEL_	*E* _Water_	*E* _PVP–CEL_	Δ*E_tot_*	Δ*E*_PVP–CEL_
1	−1,260,310	11,150.7		−1,268,520		−2940.7	
2	−1,211,520	65,544.4		−1,269,080		−7984.4	
3	−1,214,130	10,967.0	30,168.9	−1,238,560	40,947.5	−16,705.9	−188.4
4	38,279.6	11,406.2	27,575.2			−701.8	−701.8
5	−1,217,330	10,956.1	30,709.1	−1,242,320	41,609.6	−16,675.2	−55.6
6	38,594.0	11,433.8	28,054.6			−894.4	−894.4
7	−1,247,820		31,379.8	−1,263,970		−15,229.8

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
