# Peer review of "Water Effects on Molecular Adsorption of Poly(N-vinyl-2-pyrrolidone) on Cellulose Nanocrystals Surfaces: Molecular Dynamics Simulations"

_materials, 2019, doi:10.3390/ma12132155_

Round 1
Reviewer 1 Report
This manuscript, "Solvent effects on molecular adsorption of poly(N-2 vinyl-2-pyrrolidone) on cellulose nanocrystals surfaces: molecular dynamics simulations", has been well structured. Overall, the authors' explanations are logical and rational, the conclusions are convinced.
I just found some minor mistakes need to be corrected.
1) line 56, "nanopartocles" should be "nanoparticles"
2) in Figure 2. there is a missing of (d) in it. please correct it correspondingly.
3) line 249, "cellulose Iβ fragment" should be "cellulose Iβ fragment". the "β" should be subscript.
BTW, I would suggest the authors change the term "Solvent effects“ to "water effects" since only water is employed as solvent in this paper.
Author Response
1. The typo is corrected.
2. The labels in Fig. 2 are corrected.
3. The misprint is corrected.
4. The title of the manuscript is changed according to the reviewer's suggestion.
Reviewer 2 Report
At first, I thank authors for carrying out this interesting research. However, I will accept this manuscript for publishing if the authors improve the article and clarify comments below. The main revision point is to use SMD for study binding affinity in which free energy is an indicator for study interaction between PVP and CEL.
1- The author used 60 monomers for building up PVP macromolecule and 14 glucan chains (with the degree of polymerization of each chain was 10) to model Iβ cellulose. It is required to mention and reasoning the bases for the number of monomer units, chains and degree of polymerization.
2- Line 99- Please notice that the equilibrated simulation for 0.5 ns was performed in NPT ensemble.
3- Figure 2- The label of different images in Fig. 2 referred wrongly in the figure caption. Please make corrections.
4- In lines 216 to 217- This sentence “This fact also confirms our hypothesis that the polymer and cellulose mainly interact through their solvation shells.” is vague and unclear. Please illustrate how the polymer and cellulose are interacting via solvation shell using an image by which the hypothesis of solvation shell is shown.
5- Line 217 to 2018- This sentence “Moreover, a visual inspection of the trajectories shows that some water molecules form HBs with both PVP and cellulose simultaneously.” Does not support with pictures and figures. Please provide snapshots of the HBs and corresponding figures that show this assertion.
6- This study lacks scientific support since number of contacts between any pair of atoms (Fig. 7), the average number of PVP-CEL hydrogen bonds (table 2), and energy of the interaction between the systems components (table 3) cannot be an accurate indicators for evaluation of binding strengths between PVP and CEL in systems 3, 4, 5, and 6. A simple, effective, and complementary method for comparison of interaction between PVP and CEL in different conditions is to use steered molecular dynamics (SMD) simulation and calculate potential mean force (PMF) by using Jarzynski’s Equality. In this way CEL is restrained in space and PVP is pulled away from the CEL molecules by a dummy spring. Please follow the instructions in the supplement of this article and consider Figure S3 (Insulin mimetic peptide S371 folds into a helical structure) and cite this paper as a reference. https://onlinelibrary.wiley.com/doi/full/10.1002/jcc.24746
Author Response
1. The reasoning of the number of PVP monomer units, chains and degree of CNC polymerization is added in the revised manuscript.
2. The corresponding information is included in the revised manuscript.
3. The corresponding corrections are made.
4, 5. Fig. 8 in the revised manuscript illustrates PVP interactions with a cellulose fragment via their solvation shells and provides a snapshot of hydrogen bonding through water molecules.
6. We agree with the Reviewer that potential of mean force is effective method for comparison of interaction between PVP and CEL in different conditions. However, our work was aimed to estimate the possibility of PVP interaction with hydrophilic and hydrophobic facets of the CNC at a qualitative level. Moreover, since our research in this area is continuing, we going to use the instructions presented in the supplement of article “Insulin mimetic peptide S371 folds into a helical structure” to carry out SMD simulation and calculate PMF by using Jarzynski’s Equality. And probably it may be interesting task is to use Umbrella sampling method for PMF calculation additionally to this in order to compare results obtained by two ways (Jarzynski’s Equality and Umbrella). Up to now, in the literature it is discussed which method is preferable for various systems (for example, J. Chem. Phys. 128, 155104 (2008); https://doi.org/10.1063/1.2904461). Probably, such calculations for our systems may become subject of discussing of separated paper. Another reason for which we cannot supplement the article with the calculation of PMF in a short time is our modest computer resources. Despite the fact that we carry out all simulations by means of GPU, the performance for our large systems (more than 100,000 particles) is 3.080 ns/day. As for each system it is necessary to carry out at least 20–30 independent simulations in the NPT ensemble, such a numerical experiment will take a long time. In any way, we thank the Reviewer for the hint to calculate PMF and we will work in this direction.
Round 2
Reviewer 2 Report
Thanks for improving the manuscript.